# Navigation Path Based Universal Mobile Manipulator Integrated Controller (NUMMIC)

**DOI:** 10.3390/s22197369

**Published:** 2022-09-28

**Authors:** Taehyeon Kim, Myunghyun Kim, Sungwoo Yang, Donghan Kim

**Affiliations:** 1Department of Electronic Engineering, Kyung Hee University, Yongin 17104, Korea; 2AgeTech-Service Convergence Major, Department of Electronic Engineering, Kyung Hee University, Yongin 17104, Korea

**Keywords:** mobile manipulator, motion planning, simultaneous control, path analysis, ROS

## Abstract

As the demand for service robots increases, a mobile manipulator robot which can perform various tasks in a dynamic environment attracts great attention. There are some controllers that control mobile platform and manipulator arm simultaneously for efficient performance, but most of them are difficult to apply universally since they are based on only one mobile manipulator model. This lack of versatility can be a big problem because most mobile manipulator robots are made by connecting a mobile platform and manipulator from different companies. To overcome this problem, this paper proposes a simultaneous controller which can be applied not only to one model but also to various types of mobile manipulator robots. The proposed controller has three main characteristics, which are as follows: (1) establishing a pose that motion planning can be carried out in any position, avoiding obstacles and stopping in a stable manner at the target coordinates, (2) preventing the robot from collision with surrounding obstacles while driving, (3) defining a safety area where the manipulator does not hit the obstacles while driving and executing the manipulation accordingly. Our controller is fully compatible with Robot Operating System (ROS) and has been used successfully with three different types of mobile manipulator robots. In addition, we conduct motion planning experiments on five targets, each in two simulation worlds, and two motion planning scenarios using real robots in real-world environments. The result shows a significant improvement in time compared to existing control methods in various types of mobile manipulator and demonstrates that the controller works successfully in the real environment. The proposed controller is available on GitHub.

## 1. Introduction

With the development of artificial intelligence and control technology, the era has come when the service robot directly interacts with humans. The service robot is expected to have an important role in our daily life [1]. However, a current service robot is mostly only used in the same way as a mobile robot, or manipulator robot. A converged platform, in which the manipulator is loaded onto the mobile robot, is called a mobile manipulator, and since it can accomplish unstructured tasks in dynamic environments, it maximizes the use of the robots and is of especially great value in the service area [2]. This is the reason for the current research on mobile manipulators.

There are two types of mobile manipulator controllers, one for sequential control and the other for simultaneous control. Today, most of the mobile manipulator controllers in a real environment such as the field of industry use sequential control. This control method, which is shown in the Figure 1a, controls the manipulator and moves its end-effector to the goal pose after it moves the mobile robot to the manipulable location in a single goal pose. This is due not only to legal reasons, including safety concerns, but also to maintain precision and dexterity [3,4]. On the other hand, the simultaneous control method, which is shown in the Figure 1b, calculates the desired pose of a mobile robot to a single goal pose, as well as calculating how to manipulate the end-effector to reach the goal pose while the robot moves to the corresponding desired pose. After that, it executes the algorithm controlling both the mobile robot and the manipulator at the same time. Simultaneous control of the mobile manipulator has a great advantage for efficiency, and work is continuously being carried out in this area [5,6,7].

Meanwhile, because of its complexity, the typical simultaneous controller is designed to control one fixed structured mobile manipulator [8]. However, most of the mobile manipulator robots currently used do not come from one company in the form of a finished product, but are used by combining the mobile platform and the manipulator arm of different companies. This increases the possibility of each user using different mobile manipulator platforms and this situation makes it hard to apply existing simultaneous controllers built on a single model. This paper proposes the Navigation path-based Universal Mobile Manipulator Integrated Controller (NUMMIC), which is a type of kinematic controller. NUMMIC is able to control the mobile platform and manipulator simultaneously by simply changing a few parameters to overcome the existing kinematic controller’s limitations. Since most of the existing kinematic controllers need as detailed as possible characteristics of models for precise operation, it is challenging to use it with various types of mobile manipulators. However, in NUMMIC, the parameters required by the algorithm are minimized for stable simultaneous control. Accordingly, it can be applied to various types of mobile manipulators. Here, the only hardware it requires is a mobile manipulator which has a single arm on a mobile platform with Light Detection and Ranging (LiDAR) and Inertial Measurement Unit (IMU) sensor, and the software required is the Unified Robot Description Format (URDF) [9], available on ROS [10] for each part. Any mobile manipulator that satisfies these conditions can be used by the proposed controller with a few parameter revisions. This is possible because the controller performance is based on the platform’s navigation control and does not need other sensors except for LiDAR and IMU sensors during motion planning. Also, it tunes the lower-level kinematic controller rather than directly controlling the hardware at the lower end.

NUMMIC, which is suggested in this paper, is composed of three substructures: a Manipulation Enabled Pose (MEP) setting based on a LiDAR sensor, an optimized end-effector default pose, and a Manipulation Safety Section (MSS) setting based on path curvature. These substructures elaborately control the move_base package [11] and the MoveIt package [12], which are publicly provided by the ROS as mentioned, and make simultaneous control of the mobile manipulator of the target object possible in different environments and positions. Since the entire controller is operated by coordinating the lower kinematic controller packages, the controller is easily compatible with various mobile manipulators which have different specifications.

The specific structure of NUMMIC can be found in Figure 2. For a detailed explanation of the substructures, the desired pose, which is called Manipulation Enabled Pose (MEP), is needed. This pose which allows the mobile manipulator to be controlled simultaneously using global path is found by MEP setting based on LiDAR sensor. The second substructure, the optimized end-effector default pose, derives the location in which there is least deviation of time, regardless of the position of the target object and its designation as a manipulator’s default pose. This idea is based on the fact that, in the operating environment, the manipulator’s manipulation is restricted to limited directions. The last substructure, the Manipulation Safety Section (MSS) setting based on path curvature, supports the safe simultaneous control of the mobile manipulator. According to its control algorithm, manipulation is operated at the same time as the mobile platform moves. If the manipulation is being executed or has already completed its execution and the platform moves with the stretched manipulator, the possibility of colliding with obstacles near the path will become much higher. To prevent this, an algorithm from this substructure locates the area which is in no danger of colliding with nearby objects and only performs the manipulation when the mobile platform passes through the area. This area is called the Manipulation Safety Section (MSS).

In summary, this work suggests the controller, NUMMIC, for simultaneous control of the mobile manipulator. NUMMIC is able to perform in a stable manner the motion planning towards the target object in any environment when its target coordinates are given. This process takes less time compared to the sequential control algorithm, due to the simultaneous control of the mobile platform and the manipulator. Since this NUMMIC controller coordinates the move_base and MoveIt packages, which can be used publicly at ROS, it can be applied to various manipulator robots with different specifications if some parameters in the controller are modified.

To validate our proposal, we present an experiment for motion planning towards given target coordinates using the mobile manipulator robot in a simulation environment. To show the motion planning in different environments we use two different maps with various obstacles for the simulation and designate various points in the map as target coordinates. Also, to show that NUMMIC can be used in mobile manipulators with different specifications, we run the simulation with three different types of mobile manipulator: one in which UR3 [13] of universal robot is attached to Husky UGV [14] from Clearpath corporation, another one which has a UR5 [15] manipulator with the same mobile platform, and a third that has Kinova corporation’s Gen3 lite [16] manipulator on Jackal UGV [17]. Finally, we conduct an experiment using NUMMIC with a mobile manipulator, in which Husky UGV is attached to a UR3 manipulator in a real environment.

In Section 2 we explain existing approaches to mobile manipulator control. In Section 3 we describe the structure of NUMMIC in detail, including (1) its three substructures (2) the architecture based on ROS using these substructures. We present the experiments in both simulation and within a real environment, discussing the results obtained in Section 4 and Section 5. Lastly, the conclusions of the evaluation and future works are presented in Section 6.

## 2. Related Work

Because of the mobile manipulator’s capability, various related studies are conducted on the premise that it is used at different fields. The many fields of application include industry [18,19], which is mentioned above, but also construction [20], agriculture [21,22,23], disaster [24], and healthcare [25,26].

For a robot to execute complex works in these different fields and environments, it needs a highly qualified manipulation ability. In particular, the efficient control of the mobile manipulator in unstructured dynamic environments is important, but this issue is not completely solved. In Reference [27], it minimizes execution time for the pick-up task by deriving the optimal trajectory of joint space, applying a random profile approach (RPA). There are some works whose methods of control focus on the manipulator’s end-effector. [28]. One of those works carries out sampling of the waypoint with the end-effector’s target position and orientation trajectory and implements an optimized planner using the Genetic Algorithm for continuous movement of the mobile manipulator. The work in [29] efficiently generates a path without collision in complex environments by suggesting the Optimized Hierarchical Mobile Manipulator Planner (OHMP), which is composed of two steps: two-dimensional mobile motion planning and three-dimensional manipulator motion planning. Recently, due to the development of deep learning, research into mobile manipulator control using reinforcement learning is also being conducted. In [30], the authors suggest a system in which the mobile manipulator robot learns action-related places through experience-based learning with the environment. In Reference [31], they offer a mobile manipulator system with a more efficient framework by decoupling the state-of-the-art deep reinforcement learning control and visual perception.

The controller suggested in this research works as a controlling move_base, which is a package for navigation and MoveIt, the manipulator control package, on the upper level. There are some studies to find improved action by the selection in which it uses an existing controller in normal situations and replaces the motion with a new controller or adjusts the existing one in particular situations. In Reference [32], the controller normally uses move-base to drive the mobile robot and then when the robot faces the narrow space, it regenerates appropriate waypoints for passing the space. Also, in the work of Reference [33], the author suggests an algorithm which executes a more efficient exploration using Gmapping [34] and the move-base package. Moreover, in the same context, other studies are conducted which extend the function of move_base with a higher-level controller [35,36,37] or uses the MoveIt package [38,39,40] for operation in specific higher-level environments. Although it is not related to replacing or tuning the controller itself, as with reference [41], there are also some studies on user-friendly interfaces, replacing RViz [42], the 3D visualization tool which gives order to the existing move_base, and MoveIt packages, to a VR-based interface. In the case of the controllers suggested in the studies mentioned, they are activated by replacing or tuning the existing ROS packages in certain environments but have the limitation that they have to be controlled with RViz or terminals. However, in this paper, the authors make a Virtual Reality-based control interface in which the user has access intuitively at higher level. This seems consistent with the approach direction that suggests a higher-level controller which gives order to a lower one without directly handling the lower kinematic controller of the move-base and MoveIt packages.

## 3. Controller Description

The objective of the controller is to control the mobile manipulator simultaneously in various environments. For its successful operation, when the mobile manipulator receives the position of the target object, it should firstly be able to calculate the pose where it can execute the manipulation towards the target and move to the corresponding pose. The flow chart of NUMMIC for this operation can be found in Figure 3.

Following the order of the flow chart, after receiving the available position of the target, NUMMIC sets the navigation path through the global path planner [43] in the move-base package with scan data from the LiDAR sensor attached at the mobile platform. If MEP, a position where the manipulation for the target on navigation path is successfully executed, is specified, the mobile platform of the mobile manipulator can be located on the MEP controlled using move-base. This process is based on some parameters of the mobile manipulator and the information about the position of the target.

At the same time, two additional processes are required for the efficient and stable control of the mobile manipulator, which are the designation of the manipulator’s default pose and the setting for MSS using path’s curvature analysis. The designation of the suggested default pose is based on the idea that the generated trajectory at manipulation is likely to be a straight line since the mobile manipulator is aligned to face the target object when it stops at MEP. The default pose proposed in this paper is a point where, no matter what value the *z*-coordinate of the target object has, deviation of the distance from default pose’s coordinate to the target, i.e., the trajectory of end-effector during manipulation, is minimized and the manipulator does not hit the surrounding obstacles. Since the state of the manipulator not performing any action while the mobile manipulator drives is defined as the default pose, there is no part related to the calculation of default pose in NUMMIC flow chart of Figure 3. As seen in Figure 2, although the part suggesting an optimized default pose exists in parallel with other substructures, it is decided in advance by the parameters of mobile manipulator and used in NUMMIC rather than being performed in the main operation of NUMMIC. This default pose is calculated with the manipulator’s workspace and some parameters including mobile robot’s footprint, and by using it to control the manipulator it can reduce the average planning time for target objects in various locations.

The setting for MSS using path’s curvature analysis, like MEP generating algorithm, uses a navigation path to find the area with a high risk of collision between the manipulator and surroundings while the mobile robot drives. This operation can also be seen in the flow chart of Figure 3 through the *Calculate path curvature* block on the rightmost side following the *Set navigation path*. The global path generated by the global path planner from move-base is split into steps according to the granularity value, and it analyzes the curvature at each path’s interval using the difference of radian values at these steps. It distinguishes the area where there is a possibility of the mobile manipulator colliding with nearby obstacles or not depending on the analyzed curvature, the width of mobile manipulator, and the manipulation distance. Finally, it performs the manipulation only if there is MSS, the area which has no risk for the manipulator to collide with the surroundings on its path between the mobile manipulator and the target object according to the path analysis.

### 3.1. Manipulation Enabled Pose (MEP) Setting Based on LiDAR Sensor

In this part, the executed operations are as follows: (1) setting the MEP for the manipulation towards the given target and control the mobile platform for the movement to the pose, (2) stopping the MEP using the algorithm which controls the velocity proportional to distance and (3) rotation control based on P-controller to perfectly align the target and the mobile manipulator. These three operations are executed sequentially.

To set the MEP, the controller calculates the Manipulable Area. Manipulable Area is an area in which the mobile manipulator can manipulate the target. The controller then generates a global path to the target using move-base and makes the mobile manipulator drive along the path. Here, the calculation of the Manipulable Area is as follows. A pose is assumed in which the mobile manipulator stopping at MEP performs the manipulation towards the target. Figure 4 shows the pose in the simulation. Here, the first joint from the manipulator’s base link becomes the central axis of the recommended workspace. Using the characteristics of NUMMIC algorithm, only values of x and z coordinates in 3-dimensional space during the manipulation are needed since the coordinates of the base link of the mobile manipulator and the target are aligned after the control of the mobile platform. Assuming the radius of the manipulator’s recommended workspace to be $d_{m}$, *z*-coordinates of target be $t_{z}$, and *z*-coordinates of manipulator’s first joint from the ground be $m_{z}$, the angle between the line from the target, which is perpendicular to the ground, and the line extended from target to manipulator’s first joint is as follows:(1)$$\theta =\cos^{-1}\left(\frac{\left|t_{z}-m_{z}\right|\;}{d_{m}}\right)$$

The definition of the distance on the *x*-axis from the base link of the mobile platform to the first joint of the manipulator as $d_{bm}$, $r_{m}$, a radius of Manipulable Area is as follows:(2)$$r_{m}=d_{m}\sin\theta +d_{bm}$$

Thus $r_{m}$, the radius of Manipulable Area is a distance between the perpendicular foot of a point where the target meets the recommended workspace to the ground, and the perpendicular foot of a base link to the ground.

Second, the controller stops the mobile manipulator when it is determined that the mobile manipulator has entered the Manipulable Area while driving along the path. If a signal is sent instantaneously, a significant error occurrs since the controller controls the mobile robot with move-base. To prevent this, the suggested controller in this paper defines the area equivalent to twice the radius of the Manipulable Area as sigmoid distance proportional speed control area and adjusts the velocity of the mobile platform in this area. The velocity decelerates along the shape of the sigmoid function in proportion to the distance from the target point. When $v_{x}$ is the translational velocity of mobile platform, $v_{max}$ is the maximum velocity of mobile platform, $v_{stop}$ is the velocity when mobile platform is stopped, and d is the distance between the mobile platform and the MEP, then the formula of the sigmoid function is as follows.
(3)$$v_{x}=\frac{v_{max}-v_{stop}}{e^{-d}+1}+v_{stop}$$

To converge to the velocity $v_{stop}$ when it reaches the target point, the function is shifted to the positive direction as follows:(4)$$v_{x}=\frac{v_{max}-v_{stop}}{e^{-d+6}+1}+v_{stop}$$

Finally, assuming the radius of predefined sigmoid distance proportional speed control area to be $r_{sa}$, the coefficient to decelerate after entering the area is revised.
(5)$$v_{x}=\frac{v_{max}-v_{stop}}{e^{-\frac{12}{r_{sa}}d+6}+1}+v_{stop}$$

In order to verify how the mobile platform of the mobile manipulator is controlled through the above Formula (5), a graph is drawn assuming the specific situation. Because the maximum speed of Husky UGV which is used in the experiment of the paper is 1 m/s, $v_{max}$ is set at 1 m/s and the targeting velocity at stop is set at 0 m/s. Then, assuming that $r_{sa}$ about particular target A is 1.2 m, a graph of the corresponding sigmoid function is drawn as Figure 5. From this graph, it can be expected that as the mobile manipulator goes beyond the sigmoid distance proportional speed control area and approaches the target, its speed will decrease and stops in a stable manner at MEP.

After the mobile platform is stopped by the sigmoid distance proportional speed control algorithm, it rotates the platform to face toward the coordinates of target object. This rotational control is composed of simple P-control which multiplies the gain value to the difference of mobile platform’s yaw value and target one, and this gain can be modified according to the needs of the user at NUMMIC’s configuration, even though it is specified to 0.25 as default.

The final pose of the mobile manipulator becomes the MEP through a series of control processes for detailed alignment of the mobile platform including the designation of MEP by target coordinates and specification of the mobile manipulator, stop at the position, and rotational control. The graphical explanation about MEP can be found in Figure 6. Figure 6a shows where the MEP is located on the path, based on the creation of a navigation path which has the position of target object from the mobile manipulator as goal on RViz. Figure 6b shows the Manipulable Area, a circular area with radius $r_{m}$ that has the target object as a center when the mobile manipulator is located closer to the target object, and the sigmoid distance proportional speed control area with radius $2r_{m}$ which also has target object as a center. In this case, the intersection of the navigation path from the mobile manipulator to target object and the Manipulable Area becomes the position of the MEP.

### 3.2. Optimized End-Effector Default Pose

The default pose of the end-effector suggested in this paper is a pose in which there is no collision when the mobile manipulator drives, and it does not lose much time in the planning of the manipulation, whatever value the target’s *z*-coordinate has. The *z*-coordinate of the suggested pose is assumed to be k, and distance at *x*-axis from the base link of manipulator to the front footprint value on move-base of the mobile platform to be $d_{fm}$, which can be described as Figure 7. Figure 7a represents $d_{m}$, the radius of mobile manipulator’s recommended workspace, $d_{fm}$, the distance on the *x*-axis from the manipulator’s base link to the front footprint of the mobile platform, and $d_{bm}$, the distance on *x*-axis from the mobile manipulator’s base link to the manipulator arm’s base link. Figure 7b represents an arbitrary position of the target (blue point) and the expected position of default pose (green point on the green line) through a circle with the first joint from the manipulator’s base link as the origin and $d_{m}$ as the radius. A red point is the target which has a minimum *z* value among the target, and a purple point is the target which has a maximum *z* value. This range can be modified by controller’s parameter values. X, expected planning trajectory distance value from default pose to the target object, can be derived as follows:(6)$$X=\sqrt{{\left(x-d_{fm}\right)}^{2}+{\left(\sqrt{d_{m}^{2}-x^{2}}-k\right)}^{2}}$$

To find a default pose required by the controller, the formula should be expressed as the mean value and variance of continuous probability variable as follows:(7)$$E_{distance}\left(X\right)=\int_{-d_{m}}^{-d_{fm}}x\sqrt{{\left(x-d_{fm}\right)}^{2}+{\left(\sqrt{d_{m}^{2}-x^{2}}-k\right)}^{2}}\;\;dx=m$$
(8)$$V_{distance}\left(X\right)=\int_{-d_{m}}^{-d_{fm}}{\left(x-m\right)}^{2}\sqrt{{\left(x-d_{fm}\right)}^{2}+{\left(\sqrt{d_{m}^{2}-x^{2}}-k\right)}^{2}}\;\;dx$$

When the $V_{distance}\left(X\right)$ is minimum, *k* becomes the *z*-coordinate of manipulator’s default pose. Although the above expression is suitable for understanding the concept of suggested default pose, it is difficult to calculate the desired k value due to its complexity. Thus, for ease of calculation, we exchanged the values of *x* and *y*-axis, and the values in the range of integrals are replaced by discrete ones which take into consideration the manipulator’s operating range and limits in control. The revised formula is given below. Here, $z_{max}$ is the value in which the maximum height for the mobile manipulator to perform manipulation is subtracted by the *z*-coordinate value of manipulator’s first joint from the base link, and $z_{min}$ is the value in which the minimum height for manipulation is subtracted by the *z*-coordinate value of manipulator’s first joint.
(9)$$X_{2}=\sqrt{{\left(x-k\right)}^{2}+{\left(\sqrt{d_{m}^{2}-x^{2}}-d_{fm}\right)}^{2}}$$
(10)$$E_{distance}\left(X_{2}\right)=\int_{z_{min}}^{z_{max}}x\sqrt{{\left(x-k\right)}^{2}+{\left(\sqrt{d_{m}^{2}-x^{2}}-d_{fm}\right)}^{2}}\;\;dx=m_{2}$$
(11)$$V_{distance}\left(X_{2}\right)=\int_{z_{min}}^{z_{max}}{\left(x-m_{2}\right)}^{2}\sqrt{{\left(x-k\right)}^{2}+{\left(\sqrt{d_{m}^{2}-x^{2}}-d_{fm}\right)}^{2}}\;\;dx$$

As in Expressions (6)–(8), when the *V_distance_*(*X*_2_) is minimum, *k* becomes the *z*-coordinate of manipulator’s default pose $\left(z_{min}\leq k\leq z_{max}\right)$. Finally, the k value which is derived from the above calculation is designated as the *z*-coordinate of the default pose and $d_{fm}$, the distance to the front footprint of the mobile platform based on the manipulator’s base link is designated as the *x*-coordinate of the default pose. After defining the *y-*coordinate as the nearest point to 0 where the manipulator can move satisfying these two values, and inputting the orientation value so that the end-effector looks at the front of the mobile manipulator in the corresponding position, the default pose is set.

### 3.3. Manipulation Safety Section (MSS) Setting Based on Path Curvature

For successful simultaneous control of the mobile manipulator in various environments which include obstacles, it is necessary to prevent the manipulator from colliding with its surroundings. For this reason, in this paper, we analyze the global path where the mobile manipulator drives and make the manipulator decide whether there is a large curve that it might bump into when it is stretched forward along the path from the current position to the target. Assuming the radius of Manipulable Area about specific target of mobile manipulator to be $r_{m}$, the width of the mobile platform to be $w_{m}$, and the mobile manipulator to be a material particle that moves along the global path, the curvature radius ρ about the differential length of the path is required.

Generally, this curvature radius ρ, which is decided by the arc between two adjacent points of the trace, is expressed as follows:(12)$$\rho =\frac{ds}{d\theta }$$

Meanwhile, the global path from move-base is in the form of an array composed of numerous *x* and *y* coordinates. Differential length, which is called path step in this paper, can be decided using the difference between the sequential values of these *x* and *y* coordinates. Each of these path steps has a certain distance value and they are preset in advance which can be checked at the configuration of move-base package. This differential length is called path granularity, which refers to ds of above Formula (12). dθ can also be derived from the angular difference of each continuous path step. The equation for calculating the curvature radius by applying the above is as follows:(13)$$\rho =\frac{path\;granularity}{difference\;of\;path\;step\;angular}$$

Using curvature radius ρ and width of mobile platform $w_{m}$, the area where the manipulator does not collide with nearby obstacles while the mobile platform passes specific path step with its stretched arm can be calculated. This area corresponds to the space between the arc of a circle whose radius is equal to the curvature radius ρ plus half of $w_{m}$, and the arc of a circle whose radius is same as the value, ρ is subtracted by half of $w_{m}$. This is described as the green space at Figure 8. If the curvature of the path is large enough for the end-effector to be outside the green space when the manipulation is executed at that moment, the path step, which includes the base link (a green dot inside the mobile platform in Figure 8) of the mobile manipulator, is treated as not belonging to MSS. Algorithm 1 is a pseudo-code based on Python for deciding the MSS by global path.

After distinguishing the MSS and the section which is not on the global path from the mobile manipulator’s start point towards the target, if only the MSS exists on the remaining path from the path step, which is closest to the location where the mobile manipulator is driving, to the target, manipulator’s planning will be executed. Through the process, mobile manipulator can be controlled by simultaneous control without collision in complex environments including obstacles.
**Algorithm 1:** Determining the Manipulation Safety Section (MSS) from curvature of global path**Data:**Path data: path_msg [[path_msg[0]*_x_*, path_msg[0]*_y_*], [path_msg[1]*_x_*, path_msg[1]*_y_*], … [path_msg[N − 1]*_x_*, path_msg[N − 1]*_y_*]]; Path granularity: *pg*; Width of mobile robot: *w_m_*; Radius of Manipulable Area: *r_m_***Result:**Manipulation Safety Section (MSS) on the global path
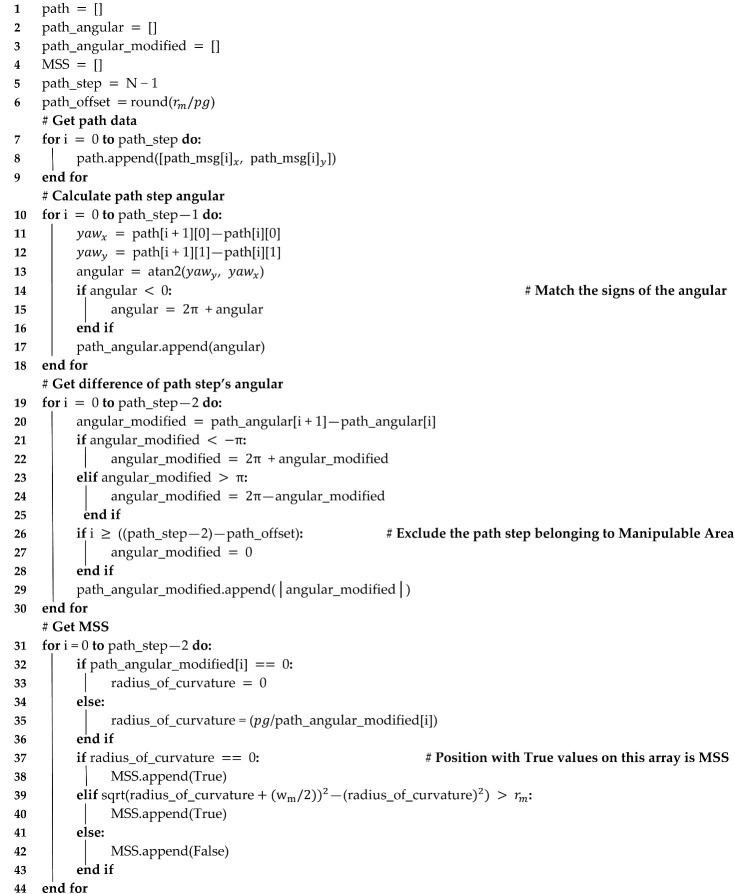


### 3.4. ROS-Based Architecture

NUMMIC is implemented through the Python scripts based on ROS, and it controls a whole mobile manipulator with move_base, which controls the mobile platform, and MoveIt, which controls the manipulator, solving inverse kinematics. The structure of NUMMIC packages on ROS is shown as Figure 9.

*Controller.launch* executes the nodes at the same time which are required for operating NUMMIC package through python scripts inside the script folder. *default_pose_cal.launch* executes the python script which calculates the position of proposed default pose using the value inside *controller_param.yaml* for the user to control the default pose that is proposed in this paper before operating NUMMIC. *controller_param.yaml* file obtains the specification of the mobile manipulator and additional setting parameters related to the operation of the controller and sends them to each node through *rospy.get_param* function. A brief description about input and additional parameters at *controller_param.yaml* file can be found in Table 1. *goal_nav.py*, *goal_nav_stop_combine.py*, and *orientation_check.py* are the Python scripts for the MEP setting, stop at MEP, and detailed rotational control after stopping. *curvature_check.py* sends the decision about MSS and its result to *manipulation.py*. *manipulation.py* executes the manipulation to locate at the proposed default pose, which is calculated in advance by the manipulator based on MSS, or at virtual goal pose, which is to reach the target coordinates for the end-effector or manipulator when the mobile platform is stopped at MEP. *default_pose_calculator.py* calculates the position of default pose which will be used with this controller according to the parameter values set by the user. Figure 10 represents how the communication using the topics works in the process in which NUMMIC controls the mobile manipulator through the external move-base package and MoveIt package on ROS. The proposed controller is released as an open-source repository on GitHub [44].

## 4. Experimental Setup

Prior to the experiment evaluating the suggested controller, we would like to explain the simulation world for the experiment, give a detailed specification of mobile manipulator, and the parameter values of NUMMIC and navigation package.

### 4.1. Simulation Worlds and Robotics Platforms

The simulation is performed for three different mobile manipulator platforms in two different worlds. Each simulation world, corresponding map files, and target coordinates are described at Figure 11. Among them, the simulation experiment in the first world involves the mobile manipulator, which always moves from the origin when motion planning for each target. Throughout this experiment, we can evaluate the mobile manipulator with the suggested controller on whether it can perform the motion planning from a fixed point to different places, avoiding the obstacles. The experiment in the second world involves the mobile manipulator that moves from MEP of previous target when it executes the motion planning for each target. Throughout the experiment, it is possible to verify that mobile manipulators existing in various locations can once again carry out motion planning towards different locations while avoiding obstacles. Also, since Test world #2 has a room-like structure in the middle, the mobile manipulator should pass through a narrow gap and exit the structure for motion planning toward Target 3 after Target 2. The experiment in such an environment is required to determine whether a collision exists during driving, which is important in a word of mobile manipulator using simultaneous control, and also it will be possible to verify the usefulness of the decision for MSS at the controller.

As mentioned previously, the mobile manipulators used for the experiments are three different types: Husky UGV with UR3 (Husky_UR3), Husky UGV with UR5 (Husky_UR5), and Jackal UGV with Kinova Gen3 lite (Jackal_Kinova). The hardware specifications of each mobile manipulator can be found in Figure 12. In addition, there are some remarks for each mobile manipulator used in the experiment. First, Husky_UR3 and Husky_UR5 share the same mobile platform but since Husky_UR3 has a bracket structure between the mobile platform and the manipulator, and Husky_UR5 does not. The height of the first joint from the manipulator’s base link is higher at Husky_UR3. Second, for the UR series provided by Universal Robots, there is a recommended reach separate from the maximum reach, but the Gen3 lite model from Kinova does not have the recommended reach described in the manual. Thus, when we used NUMMIC in the case of the UR series, the recommended reach value described in the manual was used for the parameters, but in the case of Gen3 lite, we reduced the appropriate value in the maximum reach and used it as recommended reach.

### 4.2. NUMMIC Parameters

In order to perform simultaneous control successfully for the mobile manipulator, it is necessary to put a proper parameter value according to the specifications of the mobile manipulator to be used. Although each parameter value is described sufficiently in the previous Section 3.4, this section will further explain why these values are added by the specifications. Parameters for each mobile manipulator used for experiments are represented at Table 2.

In the case of /*first_joint_height*, Husky_UR3 and Husky_UR5, using similar manipulator and the same mobile platform, should have almost same value, but as described above, Husky_UR3 has a larger value due to the difference in bracket structure. The Jackal_Kinova model has the lowest platform height but its value is higher because of the structural characteristics of the manipulator. For the/*between_base_link* parameter, Husky_UR3 and Husky_UR5 have the same value because the bracket structure mentioned earlier only affects the height, and Jacal_Kinoval’s is close to 0 since the manipulator is attached to the center of the mobile platform. The values of /*recommended_reach*,/*mobile_robot_width*, and /*base_link_offset* are input based on the hardware specification manual, and since only for Jackal_Kinova model /*recommended_reach* value is not listed on the manual, the parameter which is obtained by subtracting the arbitrary value from the maximum value is used. High /*orientation_KP_value* will reduce the time to control rotation but reduce the accuracy of motion planning, and vice versa. In this experiment, the same value is applied to all mobile manipulator models for variable control. Also, the /*refresh_cycle* value is the same for each model due to variable control, even though reducing the value can affect better driving with more system loading. /*granularity* and/*m_bl_to_ft* are written based on the parameter values in the move_base package of mobile platforms used for each mobile manipulator. Therefore, for models using the same Husky UGV, the same parameter value is applied on the NUMMIC. In the case of /*z_max* and /*z_min*, the user can enter any value as long as the mobile manipulator specification allows it, but here, the value corresponding to the limit value of the specification is added. /*z_max* is the z value that the manipulator can extend beyond the footprint of the mobile platform as much as possible, and /*z_min* is the height value of LiDAR sensor attached to the mobile platform. The reason why /*z_min* is set in this way is that if /*z_min* is smaller than the height of LiDAR, the LiDAR sensor may be covered by manipulator and interfere with driving.

### 4.3. Other Parameters

Since NUMMIC is operated at upper level of the move_base stack and MoveIt package, fine-tuning of these packages is significant for successful operation. In particular, the adjustment of parameters related to navigation plays a major role for the mobile manipulator to move successfully and efficiently to the MEP. In fact, this process should be delicately determined according to different specifications of each mobile manipulator and the configuration of the map, but in this paper, only the minimum parameter values are modified based on the default value in GitHub code supporting for each mobile platform from their companies. Because a duration of time, the time spent on Manipulation for evaluating algorithms in this paper is measured from the moment when the mobile manipulator enters the sigmoid distance proportional speed control area until the manipulation is completed. Fine-tuning of the parameter values might improve the total driving time or position error of end-effector but considering that the results of each algorithm are compared in this paper, the overall conclusion will be maintained.

Thus, we use Navfn [45] for the global path planner, and DWA planner [46] for the local path planner [47] based on navigation parameters in the repository for Husky UGV [48] and the repository for Jackal UGV [49]. *Inflation_radius* and *min_vel_x* values are modified slightly. For successful manipulation without collision at any points near the wall, the *inflation_radius* value is fixed at 0.1. Also, at Test world #2, to check whether designation of MSS is successfully carried out, the mobile manipulator should pass through the narrow gap between walls. Thus, we modify *min_vel_x* value to −0.1 and make the mobile platform temporarily move backward to pass the gap easily. The pose estimate of the mobile manipulator robot is performed by AMCL [50] that the most commonly used Monte-Carlo Localization (MCL) algorithm implemented on ROS. With the MoveIt package, the basic setting is performed based on the URDF of each manipulator arm using the supported MoveIt Setup Assistant [51]. Here, KDL Kinematics Plugin [52] was chosen for the kinematics solver.

## 5. Experimental Results

This section covers the analysis of the results of experiment for verifying our proposed controller’s performance. In Section 5.1 we analyze the results for a total of 45 times of the manipulation test with three different mobile manipulators from Target 1 to 5 in the Test world #1 simulation environment. In Section 5.2 we repeat the previous motion planning test in Test world #2 and analyze the results. Lastly, in Section 5.3, we conduct two different motion planning scenarios four times each with an actual Husky_UR3 mobile manipulator robot in real environments and analyze the results for this experiment.

### 5.1. Experiment #1: Simulated Robots in Test World #1

Table 3 shows the comparison of the average duration of time value with respect to the result of motion planning for each target using Conventional algorithms and the NUMMIC suggested in this paper. The first algorithm for comparison is the sequential controller (SC) which controls the mobile manipulator in a sequential manner. This controller executes the manipulation after its mobile platform has completely finished moving. Because it is possible to compare performance with NUMMIC when it can do motion planning for various points in the map, only the idea of MEP of NUMMIC algorithm is applied in the process of driving the mobile platform. The second algorithm, simply combined only with move_base and MoveIt (OMM), controls only the mobile platform and manipulator arm simultaneously and does not contain the velocity control which is used with NUMMIC for stability in the motion planning process and MSS decision. Duration is a measure of the time between when the mobile manipulator enters the sigmoid distance proportional speed control area and when the manipulation is completed. The reason why duration is defined in this way is that when the overall motion planning time is calculated, the duration of time varies significantly, depending on the distance from the origin to target, thus it is difficult to compare each time for different control methods. Table 4 represents the average values of the Euclidean distance error of *x*-axis, *y*-axis, and x-y plane for target coordinates and the mobile manipulator’s end-effector at the end of motion planning separately for each pre-mentioned algorithm. The reason why the error value about *z*-axis is not in the table is that the error on *z*-axis is always maintained at 0 compared to the errors on *x*-axis and *y*-axis which changed significantly by each algorithm.

The result shows that the duration was reduced by about 41% for every mobile manipulator robot when we use NUMMIC for motion planning compared to the motion planning with the existing Sequential Controller. The simultaneous control of only move_base and MoveIt can reduce the time more drastically than the method proposed in this paper, but accuracy of motion planning becomes significantly lower. In fact, the control method using only move_base and MoveIt is about +400 to +700% which is a huge surge compared to Sequential Controller. On the other hand, NUMMIC has fewer errors than that of the controller with only move_base and MoveIt, which has −12% of decreasing error or +29% increasing error with Sequential Controller.

The reason for this result seems to be closely related to which process the errors in this experiment mainly come from. Since every error is shown on the *x*-axis and the *y*-axis, not on the *z*-axis, it is estimated that the errors in the end-effector position are caused mostly by the navigation process. There are various factors for the cause of errors in the navigation process including the error from the localization between the map file and driving environment, the error by the slip of wheel, and the error that occurs when starting and stopping. These navigation errors affect the motion planning of the mobile manipulator to make errors, and in the case of OMM, which has a particularly large error, it seems that a larger error has accumulated during the stopping at the MEP. Here, it suddenly stopped without velocity control in the sigmoid distance proportional speed control area, which is one of the NUMMIC’s internal algorithm. Therefore, it takes less duration than NUMMIC, but a large slip occurs during a sudden stop and causes a large error in the end-effector position. On the other hand, in the case of NUMMIC, when entering a preset area, it stops in a stable manner at the MEP by sigmoid distance proportional speed control. This process makes a little time loss but it is still faster than Sequential Controller and, in terms of accuracy, it is not significantly different from Sequential Controller.

### 5.2. Experiment #2: Simulated Robots in Test World #2

The overall procedure is similar with previous Section 5.1. Table 5 compares the average value of duration for each target by algorithm, and Table 6 represents the average value of the Euclidean distance of the *x*-axis, *y*-axis, and *x*-*y* plane for the target coordinates and the mobile manipulator’s end-effector after motion planning is completed by each algorithm. However, there is a noticeable point which differs between Table 3 and Table 4 in Section 5.1. In the case of the control method with OMM applied to Husky_UR5, when the motion planning toward Target 3 from the position of Target 2 is executed, the manipulator always hits the wall and fails to execute. Thus, the tables below show the average value except for that result.

Through the table, the experiment conducted in the Test world #2 shows similar results to 5.1 for all mobile manipulator robots. For NUMMIC, the duration is reduced by about 38% compared to the existing Sequential Controller. And for the method with only move_base and MoveIt, the duration greatly decreases compared to NUMMIC but there is a problem that the error on the end-effector position becomes too large.

Additionally, there are several results that can be established with this experiment. One of them is the collision between the manipulator and the wall during the motion planning from Target 2 to Target 3 with Husky_UR5 and the controller with only move_base and MoveIt, which are mentioned previously. This controller simply controls the mobile platform and the manipulator arm simultaneously without any additional tuning, and it might cause serious problem in the motion planning process. In the case of the manipulator of Husky_UR3 and Jackal_Kinova, there isn’t any collision because of their short reach, but Husky_UR5’s manipulator, which has relatively long reach, can hit the obstacles. This situation can be seen in Figure 13a. In the second scene of the figure, the mobile manipulator reaches out its arm and tries to pass through a narrow gap in the third scene, resulting in a collision, which causes a completely distorted costmap, as in fourth and fifth scene. With NUMMIC, as shown in Figure 13b, this collision does not occur because it goes through the process of determining the MSS through the global path. In fact, it does not implement the manipulation because there is an area with the risk of collision on the remaining global path until the second scene of Figure 13b. However, from the third scene, which depicts the mobile robot passing a narrow gap, it starts the manipulation and successfully completes the motion planning for the target point in the fifth scene.

In addition, although the tendency for the result values of each controller is maintained, the case with Jackal UGV always shows less duration and greater error compared with Husky UGV. This result seems to be because Jackal UGV itself has a higher maximum speed than Husky UGV, and the linear and angular accelerations in [48,49] GitHub repository which are referred to in this paper are set according to this property.

### 5.3. Experiment #3: Real Robot in a Real Environment

Lastly, we conducted Experiment #3 with a real Husky_UR3 robot to check whether NUMMIC works in real-world environment without any problems. The hardware specification of Husky_UR3 and the NUMMIC’s parameter values are identical to the ones from the previous simulation. The map file of an actual environment is shown in Figure 14. This experiment is composed of two scenarios. In the first scenario, T01_Target [3, 0, 0.533] is designated as target coordinates based on the T01_Start point and the motion planning with NUMMIC is executed. Four motion planning experiments are repeated for the same point, and after each planning is completed, a laser point is applied to the end-effector to lower the foot of the perpendicular to measure which point is manipulated. In the second scenario, T02_Target [2.795, −7, 0.587] is set as target coordinates based on the T02_Start point and the motion planning with NUMMIC is done. In this scenario, the mobile manipulator must turn a corner before motion planning the target because of the structure of the hallway. In the same manner, four motion plannings are repeated for the same target, and the error between the position of end-effector and target coordinates is measured by the foot of the perpendicular in which a laser point is applied to the end-effector.

Figure 15 shows the motion planning from T01_Start to T01_Target point sequentially. Since there is no collision risk zone on the path to the target, manipulation is performed at the same time as driving (Second scene of Figure 15). After that, when it enters within a specific distance from the target point, the motion planning is successfully performed by the distance proportional speed control using sigmoid function and stopping at the MEP. Figure 16 depicts motion planning from T02_Start to T02_Target point. In this scenario, there is a high-curvature point on the path, which is an area at risk of collision, thus before passing the section, the manipulator maintains the default pose (second scene of Figure 16) and the moment it passes the section, it starts manipulation (third scene of Figure 16).

In this way, the NUMMIC operates as designed, but in terms of accuracy, the result is clearly worse than that of the simulation. Table 7 compares the average values of the error between the target coordinates and the mobile manipulator’s end-effector position after finishing each motion planning experiments: T01 and T02. In an actual experiment, since there is an uneven floor or an error on the manipulation that does not appear in the simulation, the error in the *z*-axis occurred even if it is not large compared to that in the *x*-*y* plane. Therefore, errors are also stated based on all *x*, *y*, *z* axes and Euclidean distance in three dimensions.

However, similar to the previous simulation, this result is estimated to have accumulated larger errors in the navigation process rather than derived from NUMMIC itself. In fact, prior experiments show that a significant number of errors are caused by problems from the navigation process. Since more varied non-systematic errors are easier to intervene in in real environments, the increase in errors in real environments compared to the simulation is a phenomenon that occurs in most navigation-related experiments. In the experiment of T02, which has particularly larger errors than in the simulation, most of the causes seem to have been due to the localization process between the map and the real environment, and this can be estimated in relation to the direction in which the error is large. T01 shows a relatively uniform error in the *x*-axis and *y*-axis, but the error of the *y*-axis in T02 is much larger than that of the *x*-axis. This is because in the T02 scenario the robot drives on a straight hallway and this environment lacks a point to grasp the feature, which creates a large error in the direction due to the characteristics of the localization algorithm. Thus, we can conclude that although NUMMIC operates in the same way as the simulation, due to the characteristic of the real environment, a larger error is accumulated in the navigation stage, which focuses on localization, and the manipulation stage. We expect that the errors in the entire motion planning process can also be overcome by installing additional sensors or algorithms to reduce errors from the localization stage.

## 6. Conclusions

In this paper, we proposed the NUMMIC for the mobile manipulator robot, which is able to execute in a stable manner motion planning in various environments with simultaneous control and has more time benefits than the traditional sequential control method. Here, the various environments mean that the mobile manipulator can work under various surroundings including obstacles, while also meaning that the controller can be used in different types of mobile manipulator model. What makes it possible are the structural characteristics of NUMMIC, which is composed of the designation of the MEP that selects a stable position for manipulation at different environments, the designation of the default pose position of the manipulator to help manipulation efficiently, and the safety check for the manipulator during the motion planning toward target. These three aspects control the mobile platform and the manipulator arm by tuning the move_base and MoveIt packages. Thus, any mobile manipulator platform can use this controller only if it is a mobile manipulator, the mobile platform is attached to a single manipulator arm with LiDAR and IMU sensor, and it has a URDF about the robot. This controller was tested in simulations by two different maps and three mobile manipulator platforms which have different specifications and confirmed that it operated in any environment in a stable manner. Also, in the test with real robots, the overall operation was similar to the simulation results, even though there were some errors caused by the localization process.

In the future, we will focus on reducing errors in motion planning which are inferred from the localization process and the grasping algorithm. For those, we suppose that the process for obtaining data about the grasped object using an additional camera on the wrist near the manipulator’s end effector is needed. Using NUMMIC, it will be able to reduce the error in motion planning with the method continuously checking whether it can grasp the target object at current MEP or not when the camera captures the target object while the robot moves to MEP for grasping. Also, if the grasping algorithm is added to the final process of NUMMIC using the data, this simultaneous controller for mobile manipulator can be used instantly for work such as pick and place in various environments.

## Figures and Tables

**Figure 1 sensors-22-07369-f001:**
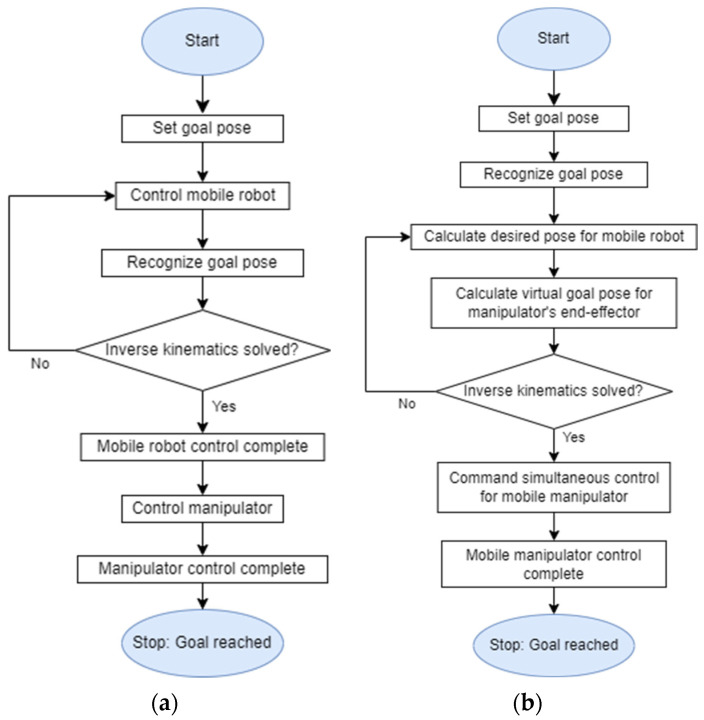
(**a**) Flowchart of the algorithm controlling mobile manipulator sequentially. (**b**) Flowchart of the algorithm controlling the mobile manipulator simultaneously.

**Figure 2 sensors-22-07369-f002:**
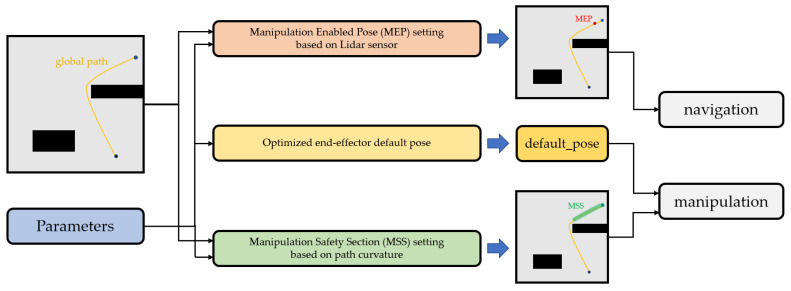
Diagram of NUMMIC’s substructures.

**Figure 3 sensors-22-07369-f003:**
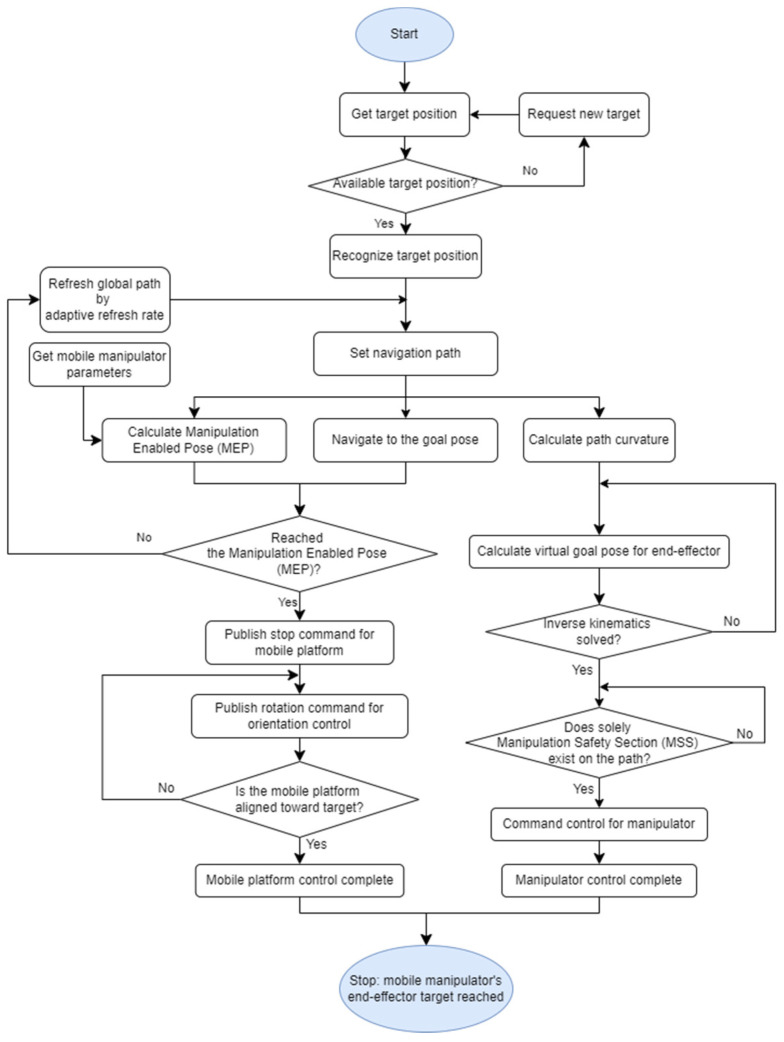
Execution flowchart of NUMMIC.

**Figure 4 sensors-22-07369-f004:**
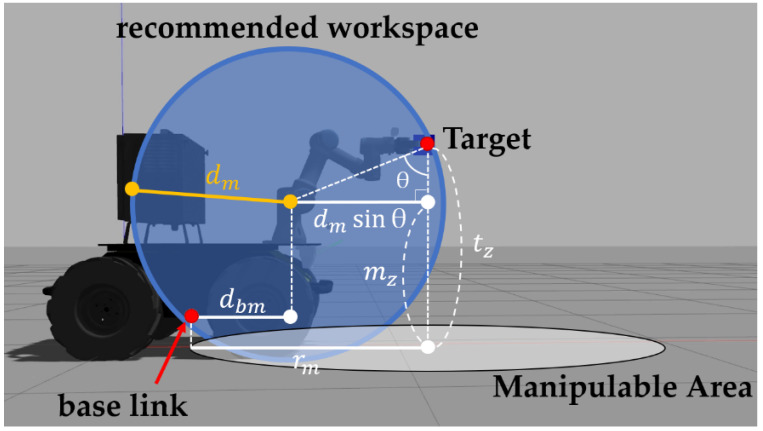
Schematic representation for the correlation between mobile manipulator’s recommended workspace, target, and *r_m_*.

**Figure 5 sensors-22-07369-f005:**
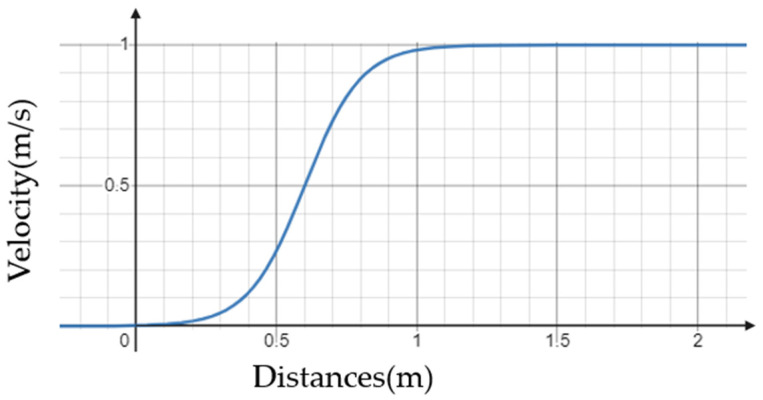
A distance-velocity graph for applying sigmoid distance proportional speed control at a point 1.2 m away from the target with a mobile platform which has a maximum speed of 1 m/s.

**Figure 6 sensors-22-07369-f006:**
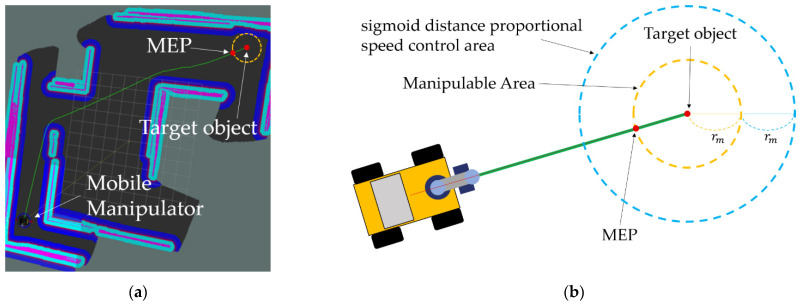
Schematic description of the Manipulation Enabled Pose (MEP). (**a**) shows where the MEP is located on the navigation path. (**b**) shows the Manipulable Area and the sigmoid distance proportional speed control area according to *r_m_*.

**Figure 7 sensors-22-07369-f007:**
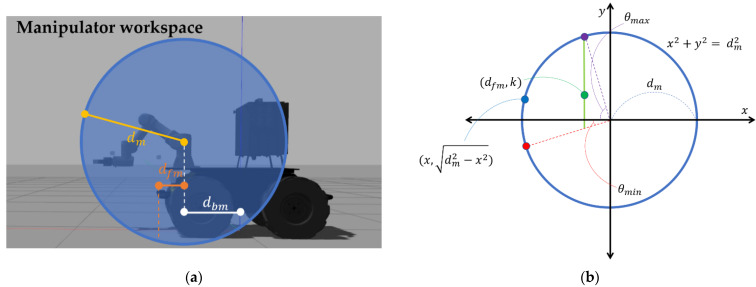
Schematic description of the position for the default pose. (**a**) represents *d_m_*, *d_fm_*, *d_bm_* on the simulation mobile manipulator model. (**b**) represents the expected position of default pose (green point) in a simplified drawing.

**Figure 8 sensors-22-07369-f008:**
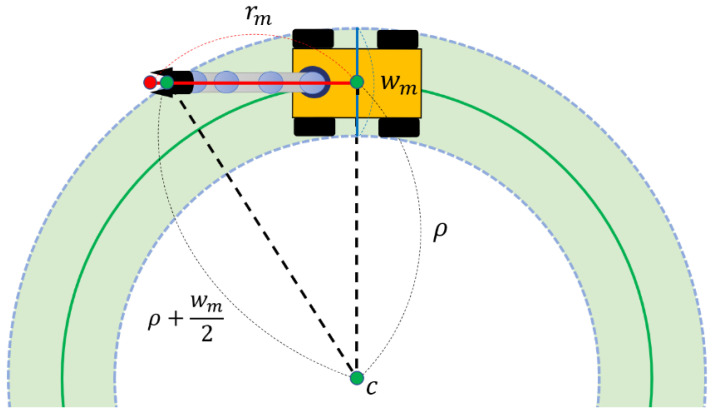
Schematic description for the calculation of Manipulation Safety Section (MSS).

**Figure 9 sensors-22-07369-f009:**
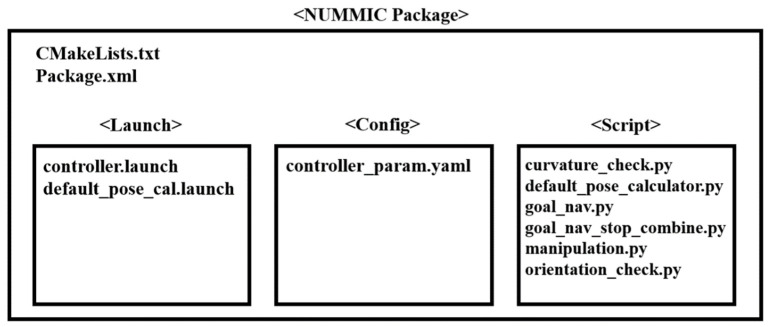
Structure of NUMMIC Package based on ROS.

**Figure 10 sensors-22-07369-f010:**
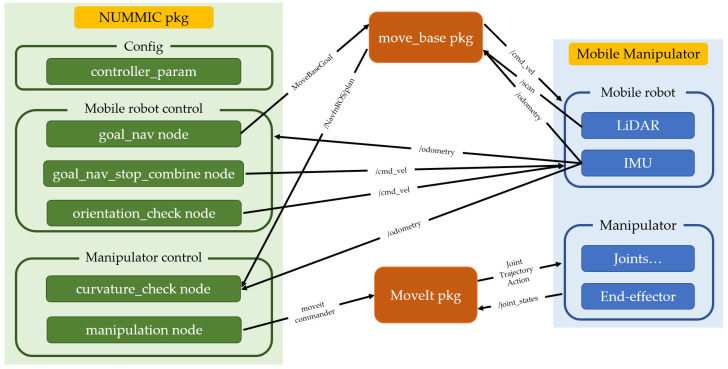
Topic exchanges between nodes in NUMMIC package based on ROS.

**Figure 11 sensors-22-07369-f011:**
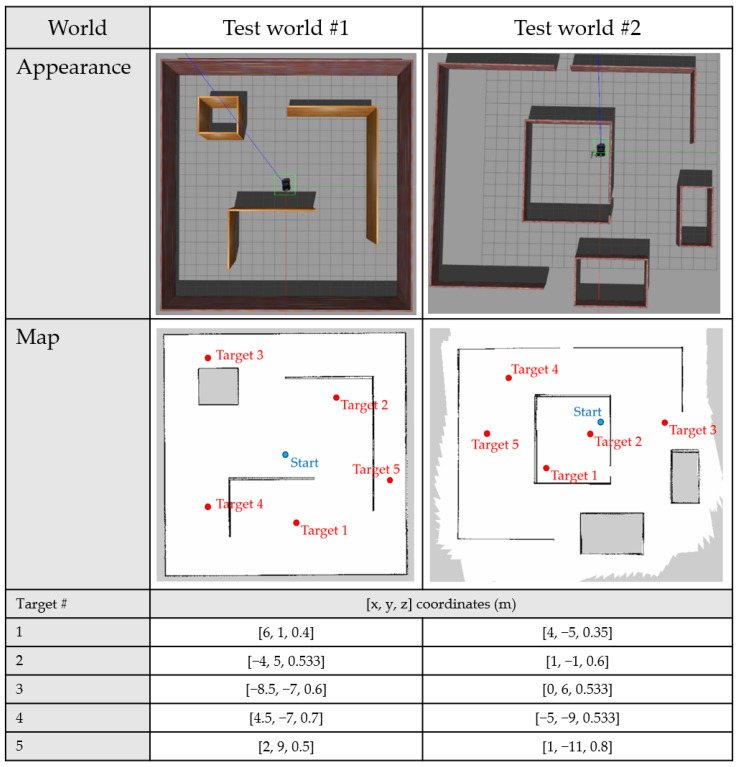
Appearance of Test worlds for each simulation tests, their map files, and [*x*, *y*, *z*] coordinates of target points.

**Figure 12 sensors-22-07369-f012:**
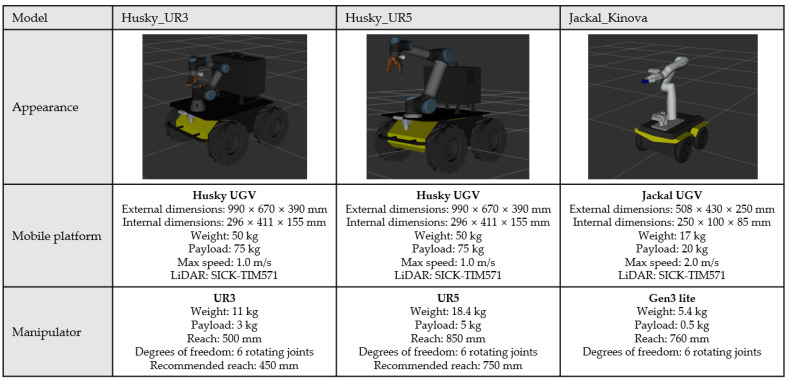
Hardware specifications of three mobile manipulators.

**Figure 13 sensors-22-07369-f013:**
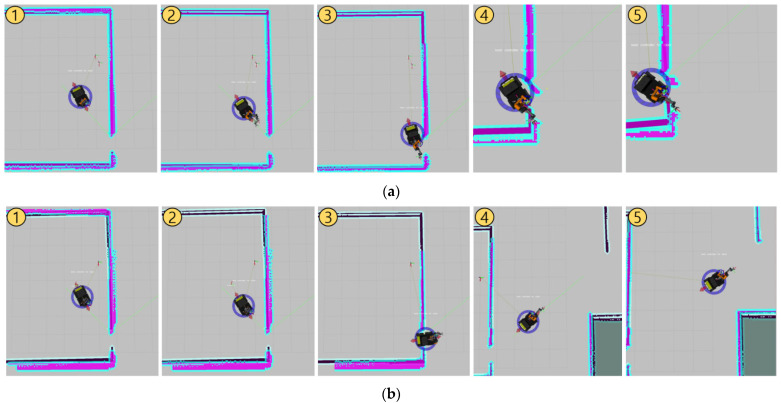
Both snapshots (**a**,**b**) show the process of motion planning from Target 2 to Target 3 in Test world #2 using Husky_UR5. (**a**) uses only move_base and MoveIt, and (**b**) uses the proposed NUMMIC controller.

**Figure 14 sensors-22-07369-f014:**
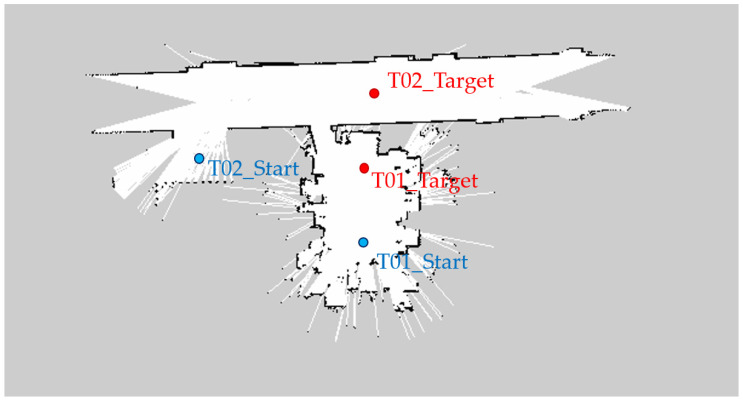
Map file of a real environment for the experiment. Each blue point, named T01_Start and T02_Start, represents the starting point of the mobile manipulator in the first and second scenarios. Each red point, named T01_Start and T02_Start, represents the target point of the mobile manipulator in the first and second scenarios.

**Figure 15 sensors-22-07369-f015:**
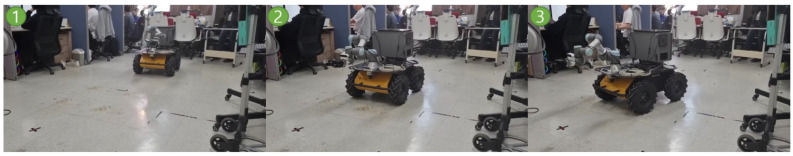
Snapshots of the experiment in the first scenario at real environment.

**Figure 16 sensors-22-07369-f016:**
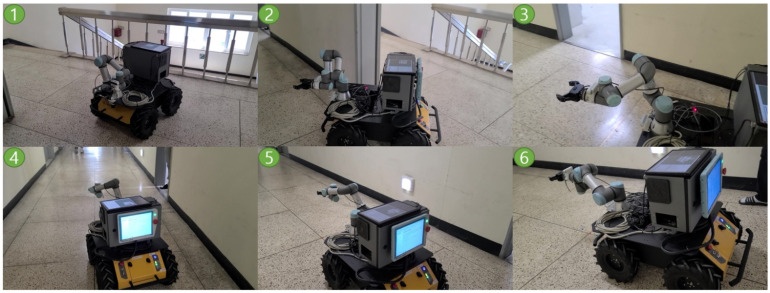
Snapshots of the experiment in the second scenario at real environment.

**Table 1 sensors-22-07369-t001:** Parameters that can be set in *controller_param.yaml* of NUMMIC Package, and brief description about them.

Parameter	Description
/first_joint_height	*z*-axis distance from ground to manipulator’s first joint (m).
/between_base_link	*x*-axis distance from mobile robot’s base_link to manipulator’s base_link (m).
/recommended_reach	Manipulator’s recommended workspace (m).
/mobile_robot_width	Width of the mobile robot (m).
/base_link_offset	Height of the mobile robot’s base_link (m).
/orientation_kP_value	kP value in orientation control.
/refresh_cycle	Global path refresh cycle (s).
/granularity	The step size to take between points on a given navigation trajectory (m).
/m_bl_to_ft	*x*-axis distance from manipulator’s base link to front footprint (m).
/z_max	Maximum z value in manipulation (m).
/z_min	Minimum z value in manipulation (m).

**Table 2 sensors-22-07369-t002:** Parameter values of NUMMIC package for each mobile manipulator used in experiments.

Parameters	Husky_UR3	Husky_UR5	Jackal_Kinova
/first_joint_height	0.533	0.474	0.52
/between_base_link	0.331	0.331	0.01
/recommended_reach	0.45	0.75	0.71
/mobile_robot_width	0.67	0.67	0.43
/base_link_offset	0.13	0.13	0.065
/orientation_KP_value	0.25	0.25	0.25
/refresh_cycle	3	3	3
/granularity	0.025	0.025	0.02
/m_bl_to_ft	0.169	0.169	0.25
/z_max	0.95	1.204	1.184
/z_min	0.312	0.312	0.25

**Table 3 sensors-22-07369-t003:** The result comparing the average values of duration using different algorithms for each mobile manipulator in Test world #1.

Average Duration of Time (S)	SC	OMM	NUMMIC
Husky_UR3	26.2	10.97	16.22
Husky_UR5	25.01	9.2	14.99
Jackal_Kinova	23.12	8.59	12.68

**Table 4 sensors-22-07369-t004:** The average values of error of end-effector after motion planning using different algorithms for each mobile manipulator in Test world #1.

Average Error (cm)	SC			OMM			NUMMIC		
	*x*	*y*	ED *	*x*	*y*	ED	*x*	*y*	ED
Husky_UR3	0.7	0.54	0.95	3.32	3.34	5.28	0.68	0.46	0.83
Husky_UR5	0.9	0.42	1.03	6.42	4.06	8.18	0.62	0.72	0.99
Jackal_Kinova	1.14	1.04	1.68	6.2	4.66	8.05	1.58	1.18	2.17

* Euclidean distance.

**Table 5 sensors-22-07369-t005:** The result comparing the average values of duration by different algorithms for each mobile manipulator in Test world #2.

Average Duration of Time (S)	SC	OMM	NUMMIC
Husky_UR3	23.92	9.98	15.25
Husky_UR5	24.73	11.37	15.5
Jackal_Kinova	22.72	8.19	13.49

**Table 6 sensors-22-07369-t006:** The average values of error of end-effector after motion planning by different algorithms for each mobile manipulator in Test world #2.

Average Error (cm)	SC			OMM			NUMMIC		
	*x*	*y*	ED *	*x*	*y*	ED	*x*	*y*	ED
Husky_UR3	0.36	0.66	0.79	5.3	5.68	8.12	0.32	0.62	0.72
Husky_UR5	0.38	0.58	0.77	1.88	2.58	3.62	0.62	0.84	1.15
Jackal_Kinova	1.7	1.52	2.48	4.22	4.86	6.77	1.52	1.08	2.14

* Euclidean distance.

**Table 7 sensors-22-07369-t007:** The error values of end-effector position after motion planning in a real environment. T01 is the result of the first scenario, and T02 is the result of the second scenario.

Average Error (cm)	*x*	*y*	*z*	Euclidean Distance
T01	2.3	2.1	0.075	3.62
T02	3.08	8.05	0.15	9.2

## Data Availability

Not applicable.

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
