# Peer review of "Navigation Path Based Universal Mobile Manipulator Integrated Controller (NUMMIC)"

_sensors, 2022, doi:10.3390/s22197369_

Round 1

Reviewer 1 Report

A very well-written and informative paper. The only issue I see in the manuscript is sometimes it is hard to follow the flow between paragraphs, especially in sections 1 and section 2. Minor English corrections are also required in the manuscript overall. 

Reviewer 2 Report

The paper identifies the lack of a versatile controller for mobile manipulators and proposes a specially designed controller able to control different types of mobile platforms and manipulators simultaneously. The approach is very interesting and allows to reach significant improvement in time also taking into account safety issues. The proposed controller has been tested in a simulation and in a real environment. 

Thus, the paper can be accepted in its current form. 

Just few suggestions in the following:

- In the title, acronyms non yet defined should be avoided;

- Symbology used in figures 4 and 7 could be recalled in the caption;

- References and acronym definition about OMM algorithm should be provided.

Reviewer 3 Report

This paper is about a novel kinematic controller for mobile manipulators. As explained in the paper, the approach can be applied to any mobile manipulator independently the DOF configuration.

Authors have presented the paper correctly, focusing on the novelty of the approach. Moreover, they have presented different experimental results as a real prove of the approach.

However, authors must improve the paper with the following revisions or comments:

-      It must be emphasized the novelty of the kinematic controller, and it must be compared with other similar approaches.

-      It must be corrected that it is not a low-level controller because it is a kinematic controller (references are poses and velocities).

-      Authors must be introduced about force and torque control of the platform because mobile manipulators must be controlled by dynamic control. The paper is focused on the kinematic control but nothing about dynamic control (torque references).
